# In Vitro and In Silico Anti-Arboviral Activities of Dihalogenated Phenolic Derivates of L-Tyrosine

**DOI:** 10.3390/molecules26113430

**Published:** 2021-06-05

**Authors:** Vanessa Loaiza-Cano, Laura Milena Monsalve-Escudero, Manuel Pastrana Restrepo, Diana Carolina Quintero-Gil, Sergio Andres Pulido Muñoz, Elkin Galeano, Wildeman Zapata, Marlen Martinez-Gutierrez

**Affiliations:** 1Grupo de Investigación en Ciencias Animales-GRICA, Facultad de Medicina Veterinaria y Zootecnia, Universidad Cooperativa de Colombia, Bucaramanga 680005, Colombia; vanessa.loaiza@udea.edu.co (V.L.-C.); lauramilemo@hotmail.com (L.M.M.-E.); dcaro63@gmail.com (D.C.Q.-G.); 2Grupo de Investigación en Productos Naturales Marinos, Universidad de Antioquia, Medellín 050001, Colombia; mhpr2017@gmail.com (M.P.R.); elkin.galeano@udea.edu.co (E.G.); 3LifeFactors Zona Franca SAS, Rionegro 054040, Colombia; spulido@lifefactors.co; 4Grupo Infettare, Facultad de Medicina, Universidad Cooperativa de Colombia, Medellín 050001, Colombia; wildeman.zapatab@campusucc.edu.co

**Keywords:** dengue virus, Zika virus, chikungunya virus, tyrosine, antiviral agents, computational biology

## Abstract

Despite the serious public health problem represented by the diseases caused by dengue (DENV), Zika (ZIKV) and chikungunya (CHIKV) viruses, there are still no specific licensed antivirals available for their treatment. Here, we examined the potential anti-arbovirus activity of ten di-halogenated compounds derived from L-tyrosine with modifications in amine and carboxyl groups. The activity of compounds on VERO cell line infection and the possible mechanism of action of the most promising compounds were evaluated. Finally, molecular docking between the compounds and viral and cellular proteins was evaluated in silico with Autodock Vina^®^, and the molecular dynamic with Gromacs^®^. Only two compounds (TDC-2M-ME and TDB-2M-ME) inhibited both ZIKV and CHIKV. Within the possible mechanism, in CHIKV, the two compounds decreased the number of genome copies and in the pre-treatment strategy the infectious viral particles. In the ZIKV model, only TDB-2M-ME inhibited the viral protein and demonstrate a virucidal effect. Moreover, in the U937 cell line infected with CHIKV, both compounds inhibited the viral protein and TDB-2M-ME inhibited the viral genome too. Finally, the in silico results showed a favorable binding energy between the compounds and the helicases of both viral models, the NSP3 of CHIKV and cellular proteins DDC and β2 adrenoreceptor.

## 1. Introduction

The emergence and re-emergence of viral diseases transmitted by dengue (DENV), Zika (ZIKV), and chikungunya (CHIKV) viruses, have generated considerable interest in public health organizations worldwide because of their high social and economic impact [1]. The transmission of these viruses is mediated by the female mosquito of the genus Aedes (arbovirus, arthropod-borne viruses), which is present in tropical and subtropical regions located at <1800 m above sea level [2]. This spatial characteristic increases the risk of transmission of ZIKV and CHIKV in the same geographic areas that have reported DENV infection [3], leading to the co-circulation of these three viruses and an increase in human co-infections [4]. Moreover, because the clinical symptoms generated during the acute stage of these viral infections are similar (principally febrile syndrome), a differential diagnosis is difficult to achieve [5], leading to an underestimation of the epidemiological data. Accordingly, it is estimated that dengue fever is perhaps the most important arbovirosis with around four billion people at risk of infection worldwide [6].

Although most of the time these three viral diseases are characterized by self-limited febrile symptoms, certain patients with dengue develop a severe form of the disease associated with hemorrhages, plasma leakage, and multiorgan damage, which can lead to death; and patients with Zika and chikungunya can develop complications such as congenic and neurodegenerative alterations [7] or chronic disease characterized by arthralgia and limited arthritis [8], respectively.

DENV and ZIKV belong to the *Flavivirus* genus and *Flaviviridae* family [9], and CHIKV belongs to the *Alphavirus* genus and the *Togaviridae* family [10]. These three viruses have a spheric morphology, low diameter, lipid envelope, and a positive-sense single-stranded RNA genome of approximately 11 kb. DENV and ZIKV present a unique open reading frame that translates into a polyprotein that after cleavage by viral and cellular proteases yields three structural proteins (capsid (C), membrane precursors (prM) and envelope protein (E)) and seven non-structural proteins (NS1, NS2a, NS2b, NS3, NS4a, NS4b, and NS5) [11]. CHIKV has two open reading frames: one produces four non-structural proteins (NSP1, NSP2, NSP3 and NSP4) and the other produces five structural proteins (capsid (C) and the glycoproteins E1, E2, E3 and 6K) [12]. In DENV, the envelope (E) protein is the antigenic determinant that defines its four serotypes (DENV1–4) [13].

The principal strategy to prevent the transmission of these viruses has been based on the campaigns of local and international organizations, aiming to stop the reproduction of the vector; however, limitations such as lack of education, reach of the campaigns, resistance to pesticides, urbanization, globalization and climatic and sociodemographic factors, has made this strategy insufficient to control transmission [14]. Therefore, the study of other strategies (vaccine development and the production of antiviral drugs) are areas of considerable world interest. The knowledge and research on these viruses and their associated pathologies through the years has provided important information allowing multinational pharmaceuticals to begin the long process of vaccine production as a mechanism to prevent infection, which can take between 5 and 18 years [15]. Currently, there is a licensed vaccine produced by Sanofi Pasteur against dengue fever (Dengvaxia^®^, CYD-TDV) that is being commercialized; nevertheless, genetic variability, age restrictions, co-circulation of the other four DENV serotypes, and the serious cases of DENV produced have complicated its market positioning. Therefore, at present, there are two vaccines in phase III of clinical trials [16]. However, the vaccines for ZIKV and CHIKV have not passed phase II of clinical trials [17,18].

Under these circumstances, effective immunization in endemic regions is still a long way off; therefore, the search and evaluation of compounds with antiviral potential that reach clinical trials to become approved drugs is a necessity. Phenolic compounds have been shown to have broad antiviral potential [19]. In addition, marine sponges have been a source for developing medications for several pathologies, including viral infections, owing to their wide pharmacological potential [20]. Moreover, considering the Colombian biodiversity and the great extension of the marine ecosystems, bromotyrosines, derived from the marine sponges *Aiolochroia crassa* and *Verongula rigida*, have been identified, and have proved to possess antiparasitic [21] and antiviral potential [22]. The limited natural sources of this type of compounds led to the use of isolated bromotyrosines structures as pharmacophores and allow the de-novo synthesis of new dihalogenated compounds derived from L-tyrosine that contain different structurally associated substitutions that already demonstrate biological and anti-parasitic activity [23]. Additionally, other tyrosine derivatives such as L-Dopa have previously demonstrated their antiarboviral activity and modulate signaling pathways related to viral replication [24,25,26]. 

Considering this information, we aimed to identify the antiviral potential (in vitro and in silico) of ten phenolic dihalogenated compounds with chlorine or bromine derived from L-tyrosine using three arboviruses of global public health importance as experimental models (DENV, ZIKV and CHIKV). According to this, we found that the antiviral activity and the possible mechanism of action of the several promising compounds is dependent on the viral model (finding activity in ZIKV and CHIKV model) and the cell line.

## 2. Results

### 2.1. Dihalogenated Compounds Derived from L-Tyrosine Are Not Toxic in the In Vitro and In Silico Models

Ten phenolic dihalogenated compounds derived from L-tyrosine were synthetized (Figure 1), and their in vitro toxicity was evaluated to identify the concentration of use of the compounds for the antiviral assays.

Based on the cytotoxicity assay, we selected a concentration of 250 µM for all the evaluations performed because none of the cultures treated with this concentration presented a viability percentage of <90% (Appendix A). For the control compounds, concentrations resulting in viabilities of >75% were selected (suramin, 500 µM; ribavirin, 200 µM; and doxycycline, 50 µM). The in silico toxicological modeling with ADMET Predictor^®^ showed that the accumulated toxicity of group I compounds (primary amines) TDC and TDB was 2 and 1, respectively. However, in group II (tertiary amines), the compounds of subgroup IIA TDC-2M-ME and TDB-2M-ME presented an accumulated toxicity of 3 and 4, respectively; however, in subgroup IIB (TDC-2M and TDB-2M), the values were 1 and 3, respectively. Finally, in group III (quaternary amines), subgroup IIIA, the toxicities of TDC-3M-ME and TDB-3M-ME were 5 and 4, respectively, and that of the subgroup IIIB compounds TDC-3M and TDB-3M were 2 and 3. TDC and TDC-2M obtained the lowest accumulated toxicity score (1 point) and TDC-3M-ME obtained the highest score (5) (Table 1).

### 2.2. Effect of Phenolic Dihalogenated Compounds Derived from L-Tyrosine on the Production of Infectious Viral Particles in the VERO Cell Line Depends on the Type of Virus

Antiviral screening showed that eight compounds significantly (*p* < 0.05) increased the number of infectious viral particles compared with the control of infection (Figure 2A, Appendix A) in cultures infected with DENV-2/S16803; the other two compounds analyzed (TDB-2M-ME and TDB-2M) had no effect and therefore were not considered a potential antiviral drug. In the ZIKV/Col infected cultures, eight compounds increased the production of infectious viral particles; however, TDC-2M-ME and TDB-2M-ME inhibited the infection (34.1% and 54.0%, respectively) (Figure 2B, Appendix A). Finally, we reported that all compounds significantly inhibited the production of infectious viral particles of CHIKV/Col (*p* < 0.05). In this sense, the percentages of infection were, for the group I compounds (primary amines), 44.5% and 33.8% (TDC and TDB, respectively); for the group II compounds (tertiary amines), 22.8%, 15.1%, 28.5% and 34.5% (TDC-2M-ME, TDB-2M-ME, TDC-2M and TDB-2M, respectively); and for group III compounds (quaternary amines), 37.5%, 50.7%, 43.5% and 39.9% (TDC-3M-ME, TDB-3M-ME, TDC-3M and TDB-3M, respectively) (Figure 2C, Appendix A).

As only two of the compounds, TDC-2M-ME and TDB-2M-ME, showed promising antiviral activity for one of the flavivirus models (ZIKV) and for the alphavirus model (CHIKV), posterior evaluations to identify possible antiviral mechanisms were only performed with these compounds.

### 2.3. TDC-2M-ME and TDB-2M-ME Only Inhibit Viral Genome Replication in the Alphavirus Infection Model Whilst TDB-2M-ME Inhibits One Viral Protein Synthesis in the ZIKV Infection Model

As a first approach to try to postulate the possible mechanism of action of the two compounds of subgroup IIA, we use different experimental methodologies. The first one was quantifying genome and viral protein in cultures with combined treatment.

In the ZIKV/Col-infected cultures, no significant differences (*p* < 0.05), where observed between the untreated control (1.06 × 10^8^ genome copies/mL) and the cultures treated with TDC-2M-ME (1.11 × 10^8^ genome copies/mL; 104.9%) or with TDB-2M-ME (9.97 × 10^7^ genome copies/mL; 94.3%). However, CHIKV/Col-infected cultures presented a statistically significant decrease (*p* < 0.05) in the number of genome copies/mL in the presence of TDC-2M-ME and TDB-2M-ME (3.15 × 10^6^ genome copies/mL; 54.5% and 2.31 × 10^6^ genome copies/mL; 40.1%, respectively) compared with the untreated control (5.77 × 10^6^ genome copies/mL) (Figure 3A).

Next, we aimed to determine if the antiviral effects of TDC-2M-ME and TDB-2M-ME were related to alterations in the translation of one viral protein for each viral model (NS1-ZIKV and E2-CHIKV) by quantification of proteins in the combined viral strategy. In the ZIKV/Col-infected cultures, TDB-2M-ME produced significant inhibition compared with the compound-free control; however, this effect was not observed with its chlorinated analog (percentage of NS1 viral protein of TDB-2M-ME, 84.5%, *p* < 0.05; TDC-2M-ME, 84.0%, *p* > 0.05). Moreover, in CHIKV/Col-infected cultures, no statistically significant differences were observed for the amount of viral protein reported in the treated cultures compared with the compound-free control (percentage E2 viral protein of TDC-2M-ME, 90.6%; TDB-2M-ME, 104.6%; *p* > 0.05) (Figure 3B).

### 2.4. TDC-2M-ME and TDB-2M-ME Treatment Prior to Infection Inhibits the Alphavirus Model (CHIKV/Col)

The second experimental methodology conducted in vitro to postulate a possible mechanism of action of the two compounds of subgroup was quantifying viral infectious particles in cultures treated with pre- and post-treatment. The pre-treatment strategy or treatment before the infection was used to identify whether the compounds have an effect on virus entry or prepare the cell to combat infection, and post-treatment strategy or treatment after the infection was used to identify if the compounds have an effect on processes after the entry of the virus (viral replicative cycle) or cellular changes related to it. In ZIKV/Col-infected and pretreated cultures were demonstrated that TDC-2M-ME significantly increased (*p* < 0.05) the number of infectious viral particles (1.27 × 10^5^ PFU/mL; 153.5%) compared with the untreated control (8.25 × 10^4^ PFU/mL) (Figure 4A). Similarly, TDB 2M-ME increased the number of infectious viral particles in the post-treatment strategy (1.46 × 10^6^ PFU/mL; 163.3%) compared with the control (8.96 × 10^5^ PFU/mL; *p* < 0.05). In the CHIKV/Col-infected cultures, the pre-treatment strategy showed that both TDC-2M-ME and TDB-2M-ME significantly decreased (*p* < 0.05) the number of infectious viral particles (7.13 × 10^7^ PFU/mL; 8.3%; 8.25 × 10^7^ PFU/mL; 9.6%, respectively) compared with the untreated control (8.25 × 10^4^ PFU/mL). However, in the post-treatment strategy, none of the compounds inhibited the infection (TDC-2M-ME 5.56 × 10^8^ PFU/mL, 74.0%; TDB-2M-ME 8.40 × 10^8^, 111.9%; untreated control 7.51 × 10^8^ PFU/mL; *p* > 0.05) (Figure 4B).

### 2.5. TDB-2M-ME Presents Virucidal Activity in the ZIKV-Infection Model

The cultures treated with TDB-2M-ME significantly reduced the number of infectious viral particles with respect to the control (6.25 × 10^4^ PFU/mL, 65.4%; 9.56 × 10^4^ PFU/mL; *p* < 0.05), indicating that TDB-2M-ME has virucidal activity. In this strategy, TDC-2M-ME did not significantly inhibit ZIKV/Col infection (7.92 × 10^4^ PFU/mL; 82.8%; *p* > 0.05). As shown in Figure 4, none of the compounds presented virucidal activity against CHIKV compared with the untreated control (TDC-2M-ME, 2.58 × 10^4^ PFU/mL, 103.9%; TDB-2M-ME, 3.01 × 10^4^ PFU/mL, 121.4%; and control, 2.48 × 10^4^ PFU/mL; *p* > 0.05, respectively) (Figure 5).

### 2.6. The Effect of TDB-2M-ME and TDC-2M-ME on the Production of Infectious Viral Particles Depends on the Cell Line

To determine if the antiviral effect observed depends on the cell line, we performed a combined strategy assay using the U937 cell line and the two compounds that previously showed antiviral activity in the previous models of viral infection. We reported that the ZIKV/Col-infected U937 cells presented a significantly higher production of infectious viral particles (*p* < 0.05) than the control (TDC-2M-ME 1.62 × 10^5^ PFU/mL, 254.6%; TDB-2M-ME 1.30 × 10^5^ PFU/mL, 204.5%; and the control, 6.36 × 10^4^ PFU/mL); however, in CHIKV/Col-infected cultures, the production of infectious viral particles was inhibited with both compounds compared with the control (TDC-2M-ME 3.59 × 10^6^ PFU/mL, 49.1%; TDB-2M-ME 5.11 × 10^6^ PFU/mL, 70.0%; and untreated control 7.31 × 10^6^ PFU/mL, respectively), as shown in Figure 6A. Considering these results, we quantified the number of genome copies and viral protein in the monolayers of the combined strategy assay, performed in U937 CHIKV-infected cells. TDB-2M-ME significantly inhibited viral genome replication in the CHIKV/Col infection model (4.52 × 10^7^ genome copies/mL, 68.1%); however, no differences were reported after treatment with TDC-2M-ME (4.90 × 10^7^ genome copies/mL, 73.8%) compared with the control (6.64 × 10^7^ genome copies/mL), as shown in Figure 6B. Moreover, as shown in Figure 6C, both compounds significantly reduced the amount of CHIKV viral proteins (percentage of viral protein 47.5%, TDC-2M-ME; and 41.4%, TDB-2M-ME; *p* < 0.05). Finally, in the pre-treatment strategy, the compounds did not inhibit infectious viral particles production (TDC-2M-ME 7.31 × 10^6^ PFU/mL, 110.4%; and TDB-2M-ME 6.88 × 10^6^ PFU/mL, 103.8%; *p* > 0.05) (Figure 6D). Similarly, in the post-treatment strategy, no inhibitory effect was found, however, TDB-2M-ME produced a significant increase of infectious viral particles (TDC-2M-ME, 6.31 × 10^6^ PFU/mL, 114.8%, *p* > 0.05; TDB-2M-ME, 8.06 × 10^6^ PFU/mL, 146.6%) (Figure 6E).

### 2.7. Phenolic Dihalogenated Compounds Derived from L-Tyrosine Present Favorable Interaction with the Viral Helicases of ZIKV and CHIKV and Some Cellular Proteins

Finally, we use in silico tools, as the last experimental methodology, to postulate a possible antiviral mechanism. This methodology identified the free binding energy between the antiviral compounds detected with the in vitro approach (TDC-2M-ME and TDB-2M-ME) and viral proteins (structural and non-structural) and proteins related to L-tyrosine derivatives.

Eight ZIKV proteins (E, fusion peptide, C, NS1, complex NS2b/NS3, NS2b/NS3 (protease domain), NS3 (helicase domain), and NS5), and five CHIKV proteins (E2, C (protease domain), NSP2 (protease domain) NSP2 (helicase domain), and NSP3 (macrodomain)) were included. The best binding energies of both compounds in the ZIKV model were reported with the NS3 protein (helicase domain), with TDC-2M-ME producing the best in silico interaction with viral proteins (−6.10 ± 0.00 kcal/mol) forming two hydrogen bonds with Arg172 and Thr261, at distances of 3.2 and 3.0 Å, and 13 hydrophobic interactions with ten amino acids followed by the interaction of TDB-2M-ME and the same protein (−5.97 ± 0.06 kcal/mol) forming three hydrogen bonds and 14 hydrophobic interactions. In the CHIKV model, the best in silico interactions of TDC-2M-ME and TDB-2M-ME were reported with NSP2 (helicase domain) (−5.80 ± 0.00 kcal/mol and −5.73 ± 0.06 kcal/mol, respectively) and NSP3 (macrodomain) (−5.67 ± 0.06 kcal/mol and −5.77 ± 0.06 kcal/mol, respectively). These compounds also presented favorable interaction energies with structural viral proteins. The free binding energies for ZIKV proteins E and fusion peptide were −4.80 ± 0.10 kcal/mol and −5.00 ± 0.00 kcal/mol for TDC-2M-ME; and −4.70 ± 0.10 kcal/mol and −4.87 ± 0.06 kcal/mol for TDB-2M-ME, respectively. The free binding energies for CHIKV E2 protein were −5.17 ± 0.12 kcal/mol for both compounds.

As we mentioned above, these compounds were evaluated with proteins related to L-Tyrosine derivatives such as DDC and four adrenergic receptors (α2a, α2b, β1 and β2). The best free binding energy of the study was obtained between β2 adrenoreceptor and TDC-2M-ME, −7.37 ± 0.06 kcal/mol, followed by this same receptor and TDB-2M-ME, −7.20 ± 0.00 kcal/mol; in both cases, the software predicted the formation of four hydrogen bonds, and the formation of 20 and 19 hydrophobic interactions, respectively. Following β2 receptor, the protein with the best interactions was DDC, with which the binding energies were −7.10 ± 0.00 kcal/mol, TDC-2M-ME, and −7.07 ± 0.15 kcal/mol, TDB-2M-ME, with the formation of two and four hydrogen bonds, and 18 and 19 hydrophobic interactions, respectively (Appendix A).

The heatmap shows the relation and differences of the binding free energies of the analyzed compounds and each crystal protein (Figure 7A). The in silico molecular interactions predicted by LigPlot^®^ software are presented in the Appendix A.

### 2.8. The Interactions between Virucidal Compound, TDB-2M-ME, and ZIKV Envelope Domains Were Not Stable over the Time

The molecular dynamic performed between TDB-2M-ME, compound with virucidal effect against ZIKV/Col, and domain III of ZIKV envelope protein had a root mean square deviation (RMSD) between 0.2 and 2.7 nm approximately (2 Å and 27 Å, respectively), during the 50 ns simulation. The largest oscillations were approximately 17 Å and were evidenced at 5 and 20 ns, while the smallest oscillations (approximately 5 Å) were evidenced between 40 and 50 ns (Figure 8A). The same compound was also evaluated with fusion peptide of ZIKV envelope protein. RMSD were between 0.2 and 9 nm approximately (2 and 90 Å, respectively), during 50 ns; but the first 20 ns of the simulation the oscillations were between 1 Å and 4 Å approximately (Figure 8B).

Then, both complexes had oscillations greater than 3 Å. With these results, none of the complexes were considered stable over the time evaluated.

## 3. Discussion

In this study, we evaluated ten synthetic dihalogenated phenolic compounds derived from L-tyrosine as antiviral candidates against three arbovirus models (DENV, ZIKV and CHIKV).

The in vitro toxicity assay showed that the viability of the compounds was >90% when cultured with the highest compound concentration (250 µM) (Appendix A), which is similar to that observed with other L-tyrosine-derived compounds such as catecholamines, L-DOPA [24] and dopamine [27]. Moreover, in the in silico assay, nine compounds presented low toxicity and only one showed medium toxicity (Table 1). This indicates that, based on their security profile, all compounds could continue in the search for possible therapeutic potential [28].

The antiviral screening performed by combined strategy showed that, in the DENV-2 model, all compounds increased the production of infectious viral particles (Figure 2A). This, seemingly pro-viral, type of response has been previously reported in this model for α-tocopherol, melatonin [29], and carbidopa [25]. This last compound inhibits the enzyme L-dopa decarboxylase (DDC), which negatively regulates flavivirus replication, leading to an increase in viral replication [25]. As this compound is a structural analog of L-tyrosine, the effect reported could be associated with the response reported in our study. Moreover, only the compounds of subgroup IIA (TDC-2M-ME and TDB-2M-ME) inhibited the production of ZIKV/Col infectious viral particles while the other eight increased their production (Figure 2B). This contrary effect in the anti-flavivirus response among structurally related compounds has been reported with carbidopa (increased viral replication as previously described) and L-DOPA (natural substrate of DDC enzyme), which decreased viral replication [25]. In the alphavirus model (CHIKV), all compounds significantly inhibited the production of infectious viral particles (Figure 2C), which makes this the first study reporting anti-alphavirus activity of L-tyrosine derived compounds. Finally, the percentages of inhibition of TDC-2M-ME and TDB-2M-ME in the CHIKV model were higher than in the ZIKV model. The differential effects observed in the arbovirus models using the same antiviral strategy has been reported for small molecules compounds such as orlistat, which inhibited the in vitro infection of two CHIKV isolates but only one of the ZIKV isolates [30], and in lupeol acetate and voacangine whose efficiency was ten times higher in the DENV infection model than in the CHIKV model [31].

Furthermore, we try to postulate the inhibitory mechanisms in vitro of the two most promising compounds studied (TDC-2M-ME and TDB-2M-ME), in both infection models (ZIKV and CHIKV), assessing the combined strategy and measuring the replication of the viral genome and the synthesis of two viral proteins. In this sense, both TDC-2M-ME and TDB-2M-ME inhibited genome replication only in the CHIKV/Col infection (Figure 3A), and TBD-2M-ME affected NS1 viral protein production in the ZIKV/Col model (Figure 3B). Overall, these results could be associated with existing differences in the formation of the replication complex and the alteration of cellular processes participating in this complex, in alphavirus and flavivirus [32]. Therefore, we suggest that the antiviral effect observed in the CHIKV model is related to a reduction in genome replication, which in turn would affect the release of infectious viral particles. In the ZIKV model, despite being analog compounds, the antiviral effect of TDB-2M-ME would be related to the production of antiviral proteins; however, the effect of TDC-2M-ME would occur after viral genome replication and protein synthesis. Additional studies will be necessary to identify in detail these mechanisms.

The pre-and post-treatment strategies aimed to elucidate if the antiviral effect occurs in previous and early stages of infection or after the virus enters the cell. We reported that in ZIKV/Col-infected cultures, TDC-2M-ME significantly increased the production of viral particles in the pre-treatment strategy and that TDB-2M-ME had the same effect in the post-treatment strategy (Figure 4). Considering this virus induces endoplasmic reticulum (ER) stress and activation of unfolded-protein-response (UPR) pathways, which leads to the generation of autophagosomes and consequent enhancement of replication [33], and that other L-tyrosine-derived compounds, such as catecholamines, can induce ER stress resulting in changes in the response to the UPR pathways, which leads to autophagy activation in response to stress [34]; could be possible that either TDC-2M-ME or the infection per se have a synergistic effect that favors the increase in infection. In CHIKV/Col-infected cultures, both TDC-2M-ME and TDB-2M-ME inhibited the production of infectious viral particles in the pre-treatment strategy; however, neither of the compounds showed an effect in the post-treatment strategy (Figure 4). As it has been reported that an inducer of ER stress (thapsigargin) diminished the production of infectious viral particles in the CHIKV/Col infection model [35], and the compounds analyzed in our study are possible ER stress inducers, it is feasible to consider that these compounds could be inhibiting CHIKV/Col infection.

On the other hand, only TDB-2M-ME presented a virucidal effect in the ZIKV model by decreasing the production of infectious viral particles (Figure 5). Virucidal activity has been associated with complex molecules such as proanthocyanidin [36]; nevertheless, in the Influenza A model, it has been described that small esterified molecules such as oseltamivir and zanamivir [37] possess virucidal activity through direct binding with the structural protein neuraminidase. Therefore, because TDB-2M-ME is a small, esterified molecule, its virucidal activity could be related to binding to a viral protein involved in viral adhesion with the receptor of the host. Furthermore, the fact that the halogenated compound containing bromine showed virucidal activity and the compound containing chlorine did not (TDC-2M-ME) could be associated with the binding strength of bromine, which is higher than that of chlorine [38,39]. Nevertheless, no virucidal activity was observed in the CHIKV/Col infection model, which might be explained by the different conformation of the structural proteins of the flavivirus and alphavirus [40].

In the last in vitro assay, we evaluated the antiviral activity of the compounds of subgroup IIA (TDC-2M-ME and TDB-2M-ME) in monocytes/macrophages (U937) due to their involvement in the pathogenesis of these viral models [41]. Unlike the results obtained in VERO cultures, U937 cells infected with ZIKV/Col enhanced the viral infection (Figure 6A). This cell line-dependent effect has been reported in cultures treated with sofosbuvir and infected with ZIKV, in which an antiviral effect was detected only in Huh7 cells but not in VERO and A549 cell lines [42]. In the CHIKV/Col model, both compounds inhibited the production of viral particles (Figure 6A); for TDB-2M-ME, this inhibition would be dependent on the replication (Figure 6B), and, unlike that found in VERO cells, for both compounds, the inhibition would be associated with the viral protein (Figure 6C), because in the pre-and post-treatment strategies there was no decrease in infection (Figure 6D–E). These variations in activity could be related in the first place, to protein reprogramming and signaling pathways induced in the viral infection that may be dependent on the cell line [33,43,44]; and second to the cellular metabolome, which changes depending on the arboviral infection, as each type of virus produces different metabolites [45], leading to differences in the metabolisms of the compounds in each cell type.

In addition to the in vitro results, in silico information could help to propose a possible mechanism of action of compounds with antiviral activity, principally those that could directly act on viral proteins and/or cellular proteins related with L-tyrosine derived molecules. Therefore, molecular docking was used to analyze the compound-protein interactions and to generate the heat map (Figure 7).

The anti-ZIKV effect of compounds in the combined strategy could be attributed to the predicted interaction with motifs I and V of domains 1 and 2 of the protein NS3 because the binding energies were the most favorable in this model. Further, taking into account that these motifs have a helicase function and are related to the binding and hydrolysis of NTPs [46], this could be considered as a possible antiviral mechanism of compounds analyzed in our study. The favorable free binding energies obtained between the compounds and the DDC and adrenergic receptors must also be considered, which would be aligned with the possible mechanism of action proposed in previous paragraphs (Figure 7). Unlike our expectations, TDB-2M-ME (the only compound with virucidal activity against ZIKV/Col) had one of the least favorable free binding energies of all the proteins evaluated with protein E (Figure 7), and the complexes evaluated by molecular dynamics were not stable for 50 ns (Figure 8), only for 20 ns in the complex with envelope-fusion peptide (Figure 8B). This could suggest that TDB-2M-ME does not act directly on viral structural proteins evaluated (domain III of protein E and fusion peptide) like other small-molecules but could do it in other domains or by a different mechanism [47] complemented with the short union to the fusion peptide. These results could indicate that the compounds possibly have a direct synergistic activity on the structural and non-structural viral proteins and an indirect activity on cellular mechanisms in the uninterrupted presence of the compound. Other experimental strategies to identify the potential enzymatic inhibition must be conducted to prove the prediction obtained with in silico tools.

For the CHIKV model, favorable binding energies were observed between TDC-2M-ME and NSP2 (helicase domain) and between TDB-2M-ME and NSP3 (macrodomain), despite forming hydrogen bonds with a distance >3.0 Å. It has been reported that a distance of <2.9 Å between the donor group and the acceptor atom is favorable for hydrogen bond formation [48], and therefore is likely that the favorable nature of the interaction between the compounds (TDC-2M-ME and TDB-2M-ME) and the viral proteins is primarily caused by hydrophobic interactions rather than the hydrogen bonds (Figure 6, Appendix A). These in silico results could explain the decrease in genome copies in the CHIKV/Col cultures treated with TDC-2M-ME and TDB-2M-ME (Figure 3A) because it has been described that NSP3 inhibition truncates viral replication [49] and that helicase function is essential to this process [32]. Future in vitro evaluations will help to further elucidate these hypotheses.

The results demonstrated that the antiviral activity of the dihalogenated compounds derived from L-tyrosine depends on the arbovirus model used. Moreover, the only compounds with antiviral activity against both ZIKV and CHIKV were TDC-2M-ME and TDB-2M-ME, and their possible mechanisms of action were dependent on the arboviral model and cell line. Finally, the use of computation tools to help to postulate the potential mechanisms of action of antiviral compounds and predict their toxicological response leads us to a new era of development and rational evaluation of molecules with drug potential. Future studies evaluating the cellular processes that could be affected by these compounds could provide answers on the possible host-directed mechanisms of action that have been proposed.

## 4. Materials and Methods

### 4.1. Synthesis of Phenolic Dihalogenated Compounds Derived from L-Tyrosine

Synthesis and dihalogenation were performed with substitutions in the phenolic ring at positions 3 and 5. The molecular structure was characterized by ^1^H and ^13^C nuclear magnetic resonance spectroscopy and high-resolution mass spectrometry [23]. The ten compounds were classified into three groups as per the substitution in their amine group (Figure 1).

### 4.2. Cells, Viruses and Controls

The assays were performed in VERO (ATCC^®^ CCL-81™; derived from African green monkey kidney epithelial cells, *Cercopithecus aethiops*) and U937 (monocytes, adherent clone) cell lines. The cells were cultured in DMEM (Dulbecco’s Modified Eagle Medium, GIBCO^®^, Grand Island, NY, USA) supplemented with 2% fetal bovine serum (FBS, GIBCO^®^) and 1% antibiotic/antifungal (streptomycin 10 mg/mL, penicillin 10,000 U/mL and amphotericin B 0.025 mg/mL, GIBCO^®^), and kept in a humid atmosphere containing 5% CO_2_ at 37 °C. For the antiviral assays, was used a reference strain for DENV (DENV-2/S16803), and two Colombian clinical isolates for ZIKV and CHIKV (VERO and U937 cells were infected with an MOI of 5 and 1, respectively). CHIKV/Col had been previously reported [31], whereas ZIKV/Col was isolated from serum from a patient infected during the 2015 epidemic in Colombia [4,50]. This ZIKV/Col isolation was obtained from the inoculation of C6/36 cells. Identification was performed on the supernatants by conventional polymerase chain reaction (PCR) using the previously described primers ZIKVF9027-ZIKVR9197c [51] that flank a 367-bp envelope fragment and quantitative polymerase chain reaction (qPCR) was performed using primers 1086 and 1162c [52] with a 1107-FAB probe that targeted a region of the envelope protein. A fragment from a partial region of the envelope was amplified using the previously reported primers ZIKVENF and ZIKVENVR [53] and the obtained amplicon was sent to Macrogen Inc. (Seoul, Korea) for sequencing with the Sanger method (Appendix A). Moreover, previously reported positive controls of inhibition (suramin, ribavirin and doxycycline, Sigma-Aldrich^®^, SigmaAldrich Chemical Co., St. Louis, MO, USA) were used in all cases depending on the viral model and antiviral strategy [54,55,56,57].

### 4.3. Toxicity Assay

#### 4.3.1. Cytotoxicity

The cytotoxicity of the compounds, controls and their solvents were evaluated using MTT assay (thiazolyl blue tetrazolium bromide, Sigma-Aldrich^®^). First, 3 × 104 cells (VERO or U937) were seeded in 96-well plates and, 24 h later, treated with seven serial dilutions of each compound (7.8–500 µM) and six of each control (suramin, 15.6–500 µM; ribavirin, 6.3–200 µM; and doxycycline 6.3–200 µM). After 48 h, the supernatants were discarded, the monolayers of cells were washed with PBS, and MTT (0.5 mg/mL) was added. The plates were then incubated for 2 h; and, following the dissolution of the DMSO crystals, the absorbance was read at 450 nm. Viability percentage was calculated based on the absorbance of the controls without the analyzed compounds (100% viability). Each compound was then evaluated by triplicate in two independent experiments (*n*: 6). Toxicity was assessed at only one concentration in the U937 cell line (compounds: 250 µM; ribavirin: 100 µM; and doxycycline: 50 µM). Subsequent assays were performed using only one concentration identified as non-toxic (viability ≥80%).

#### 4.3.2. In Silico Toxicological Modeling

The in silico toxicological modeling of compounds was conducted using ADMET Predictor^®^ v8 software from Simulation Plus (www.simulations-plus accessed on 19 September 2018) and molecular structures were modeled with ACD/ChemSketch^®^ 12.01 (Freeware Version). ADMET Predictor^®^, is a toxicity simulator based on multivariate statistical models applied to known QSAR data, for the creation of toxicity prediction algorithms in different models. The predictions of the programs have adjustments in their biological models ranging from 55% to 95%. The information obtained with these biocomputational tools helps to make toxicological decisions that allow replacing, refining and reducing the work with laboratory animals. The potential toxicity level of each compound was defined on a scale from 0 to 13 (according to a number of qualitatively evaluated parameters); if the compound was toxic for any of the parameters, one point was assigned. The results were reported as accumulated toxicity and the compounds were classified, depending on the number of points, as compounds with low toxicity (1–4 positive parameters), medium toxicity (5–8 positive parameters), or high toxicity (9–13 parameters).

### 4.4. In Vitro Antiviral Strategies

To identify the most promising antiviral compounds, first, a screening was conducted in VERO cells using the three viral models. In the combined treatment strategy, 6 × 10^4^ cells per well were plated into 48-well plates; 24 h later, the cultured cells were treated with each compound. After 24 h, the supernatant was discarded and a mixture containing equal parts of compound and virus (each of the compounds mixed with each of the viruses) was used to inoculate the monolayers of cells for 2 h. Then, the viral inoculum was removed and fresh compound was added. Moreover, 24 h post-infection, the supernatants and monolayers of cells were collected and stored at −70 °C until they were used for titration using a plaque assay.

The same combined treatment strategy was later used to postulate the possible antiviral action mechanisms of the most promising compounds. For this purpose, the collected monolayers were analyzed by RT-qPCR; and the cells that were seeded in 96-well plates (3 × 10^4^ cells/well), treated, fixed with 4% PFA and stored in PBS at 4 °C, were processed by Cell-ELISA. Moreover, two other experimental strategies were implemented. In the pre-treatment strategy, the cells were treated for 24 h with the compounds and later inoculated with each of the viruses; the inoculum was left for 2 h and then removed. In the post-treatment strategy, the cells were inoculated with the virus; after 2 h of incubation, the inoculum was removed, and the cells were treated with the compounds for 24 h. Both in pre and post-treatment strategies, the supernatants were collected 24 h post-infection and stored at −70 °C until they were used for titration using a plaque assay.

Finally, to identify possible virucidal effects, a compound/virus (1:1) mixture was prepared and pre-incubated for 1 h at 4 °C before being used to inoculate the cultures for 2 h. Then, the inoculum was removed and 1.5% carboxymethyl cellulose (CMC) was added. The cells were then fixed for counting as will be described later.

### 4.5. Quantification of Infection

Infection was quantified by three different methodologies: titration by plaque assay, RT-qPCR, and Cell-ELISA. In each case, the percentage of infection was calculated based on the untreated control (100% infection) as per the units of measure of each technique.

#### 4.5.1. Titration by Plaque Formation

In this assay, 1 × 10^6^ VERO cells per well (24-well plate) were inoculated with serial dilutions of supernatants. After 2 h, the inoculum was removed and 1.5% CMC was added. CMC was then removed at days 4, 7 or 12 depending on the virus (CHIKV/Col, ZIKV/Col, and DENV-2/S16803, respectively), and the cells were then fixed with 4% PFA and stained with crystal violet. The plaques were then counted and the results were expressed as plaque-forming units per milliliter (PFU/mL).

#### 4.5.2. RT-qPCR

For the absolute quantification of the genomic copies, 76 pb of E and 125 pb of NSP4 of ZIKV/Col and CHIKV/Col were cloned, respectively [52]. The amplified PCR products were ligated into the pJET1.2/blunt vector and cloned using the CloneJET kit (Thermo Scientific, Waltham, MA, USA), following the previously described protocol [58,59]. Total viral RNA extraction from the cell monolayers was conducted with the ZYMO^®^ Quick-RNA™ Viral kit, retrotranscription (from 500–1000 ng of RNA) with the High-Capacity cDNA Reverse Transcription Kit (Thermo Scientific^®^), and amplification with the Power Up™ SYBR Green Master Mix kit (Thermo Scientific^®^), following the manufacturers’ instructions. The samples were amplified in a QuantStudio 3 thermocycler (Thermo Scientific^®^); the thermal profile used was 1 cycle at 50 °C for 2 min, 1 cycle at 95 °C for 2 min, 40 cycles at 95 °C for 15 s and 60 °C for 1 min.

#### 4.5.3. Cell ELISA

The cell monolayers fixed with 4% PFA were treated with 0.1% Triton X-100. Then, 0.3% H_2_O_2_ in 10% methanol in PBS was added and later the non-specific binding sites were blocked with 10% BFS-PBS. The cells were incubated overnight at 4 °C with anti-E2-CHIKV protein or anti-NS1-ZIKV protein (clone A54Q and clone EA88, respectively; Thermo Fisher Scientific^®^) mouse monoclonal antibody (Ac). Then, they were incubated with an anti-mouse-HRP secondary antibody for 30 min at room temperature. Finally, 3,3′,5,5′-tetramethylbenzidine (TMB; Invitrogen^®^, Carlsbad, CA, USA) was added. The absorbance was read at 620 nm using a Multiskan™ FC Microplate Photometer (Thermo Scientific^®^) reader.

### 4.6. Statistical Analysis

At least two independent experiments with two repetitions (*n*: 4) were conducted for each of the antiviral strategies. Each experiment included untreated cultures (controls of infection) and cultures treated with the inhibition controls suramin, ribavirin, doxycycline, and/or UV light (controls of inhibition). To identify differences between experimental groups and the untreated control, the Student’s *t* parametric test was used with values lower than 0.05 (*p* < 0.05) considered statistically significant.

### 4.7. In Silico Assays

The interactions of compounds with viral proteins were evaluated by molecular docking and molecular dynamic. The structure of potential antiviral compounds was obtained with ACD/ChemSketch^®^ 12.01 (Freeware Version) software. For the molecular docking analysis, the tridimensional structures of 13 viral and five cellular proteins (with a resolution equal or lower than 3.0 Å) were downloaded from the Protein Data Bank (PDB) database. To determine possible compound-protein binding sites, we used the PeptiMap tool in some cases [60], whereas in others we performed a bibliographic search of biologically relevant sites. The 3D models and grid box were prepared for docking using Python Molecular Viewer (PMV) [61], as previously described [62]. Finally, molecular docking in the defined coordinates of grid box was evaluated with an exhaustiveness value of 10 in Autodock Vina [63]. The binding free energies obtained were used for the development of a heatmap. Moreover, the evaluation of possible interactions was performed using PMV [61] and LigPlot^®^ [64]. The molecular dynamic was performed with the molecular complex formed by two domains of ZIKV viral protein E and the compound with a virucidal activity for 50 ns using the GROningen MAchine for Chemical Simulations (GROMACS 5.1.4, http://gromacs.org (accessed on 8 September 2020)) [65,66], as previously described [60].

## Figures and Tables

**Figure 1 molecules-26-03430-f001:**
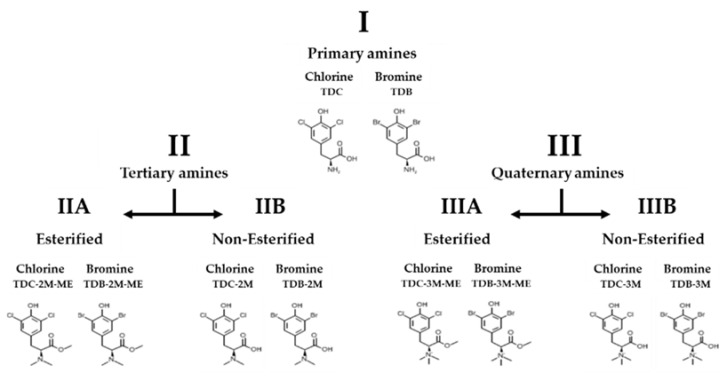
Classification of evaluated compounds. Ten phenolic dihalogenated compounds derived from L-tyrosine were synthetized, then were classified into three groups as per the substitution in their amine group: I (Primary amines), II (Tertiary amines) and III (Quaternary amines). Finally, groups II and III were divided into sub-groups according to the carboxyl substitution with a methyl group (A or B, with or without esterification, respectively). Each sub-group has one di-chlorinated compound and one di-brominated compound.

**Figure 2 molecules-26-03430-f002:**
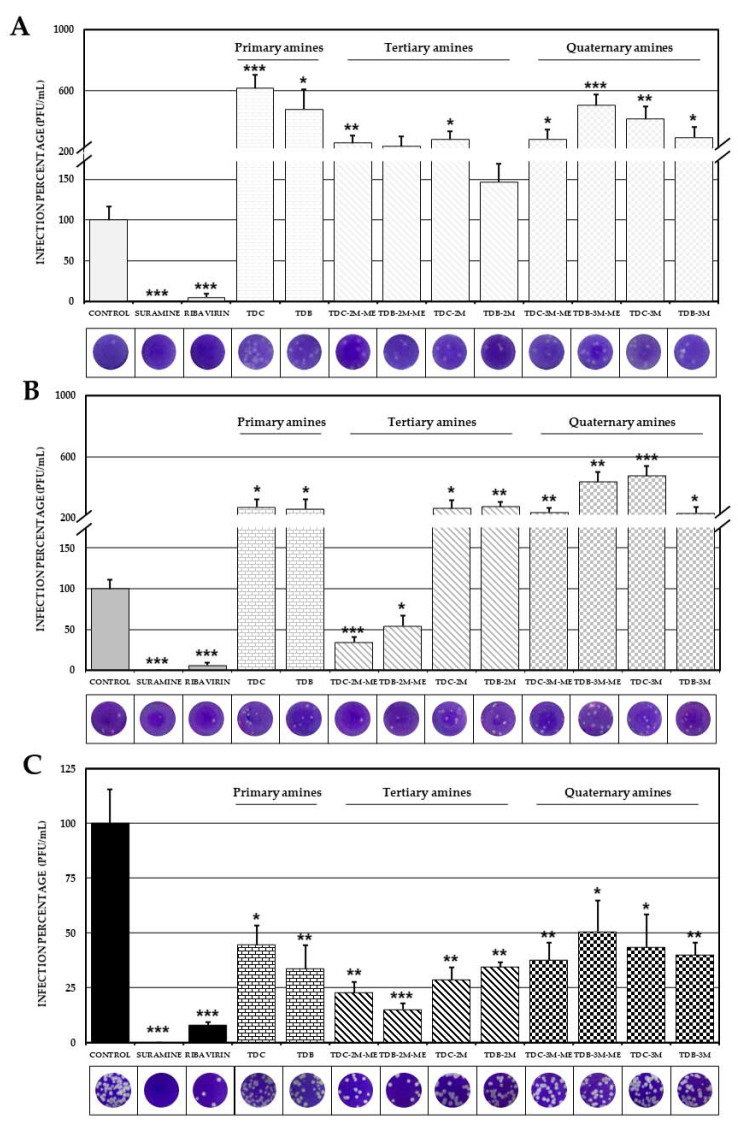
Effect of compounds on the production of infectious viral particles. Percentages of infection calculated according to the results obtained by plaque assay (PFU/ml) of the supernatants collected from the antiviral screening of the compounds in VERO cells infected with DENV-2/S16803 (**A**), ZIKV/Col (**B**) or CHIKV/Col (**C**). In all cases, the control without treatment was assumed as 100% infection. The asterisks indicate statistically significant differences with respect to the control without compound (* *p* < 0.05, ** *p* < 0.01 and *** *p* < 0.001; *t*-Student) and error bars indicate standard error of the mean; *n*: 6. Moreover, are shown representative plaques of titration on VERO cells corresponding to each experimental condition.

**Figure 3 molecules-26-03430-f003:**
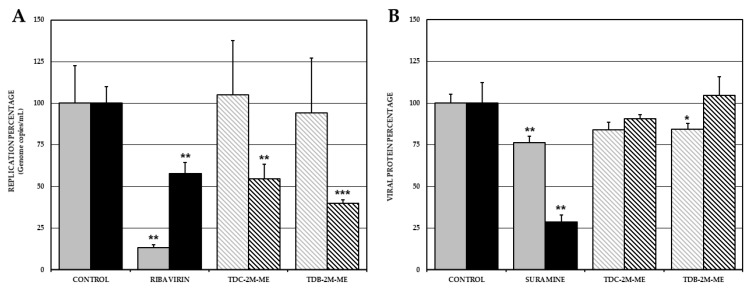
Effect of the compounds TDC-2M-ME and TDB-2M-ME on viral replication and translation in VERO cells. Percentage of viral genome replication obtained by real-time PCR (genome copies/mL) in VERO cells infected with ZIKV/Col (gray bars) or CHIKV/Col (black bars) at MOI 5, assuming the control without treatment as 100% infection (**A**). Percentage of viral protein of ZIKV/Col (gray) and CHIKV/Col (black) evaluated by Cell-ELISA (absorbance) (**B**). The asterisks indicate statistically significant differences with respect to the control without compound (* *p* <0.05, ** *p* <0.01 and *** *p* <0.001; *t*-Student) and error bars indicate standard error of the mean; *n:* 4.

**Figure 4 molecules-26-03430-f004:**
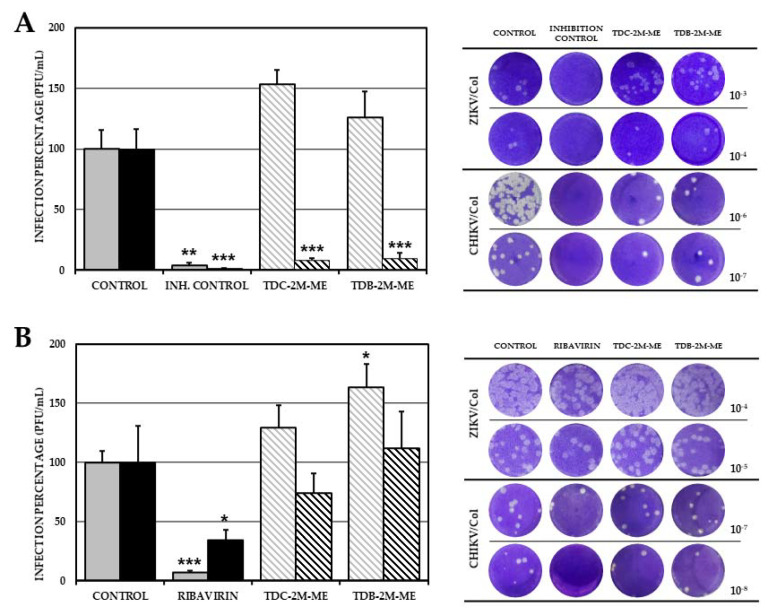
Effect of the compounds TDC-2M-ME and TDB-2M-ME on the production of infectious viral particles according to the treatment strategy in VERO cells. Percentages of infection calculated according to the results obtained by plaque assay of the supernatants collected from VERO cells treated before (**A**) or after (**B**) the infection with ZIKV/Col (gray bars) or CHIKV/Col (black bars). The control without treatment was assumed as 100% infection. The asterisks indicate statistically significant differences with respect to the control without compound (* *p* < 0.05, ** *p* <0.01 and *** *p* <0.001; *t*-Student) and error bars indicate standard error of the mean; *n:* 6. Moreover, are shown representative plaques of titration on VERO cells corresponding to each experimental condition.

**Figure 5 molecules-26-03430-f005:**
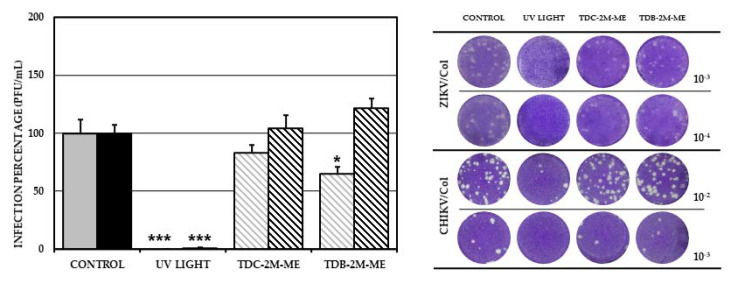
Virucidal effect of the compounds TDC-2M-ME and TDB-2M-ME on the production of infectious viral particles. Percentages of infection calculated according to the results obtained by plaque assay of VERO cells treated and simultaneously infected with ZIKV/Col (gray bars) or CHIKV/Col (black bars). The control without treatment was assumed as 100% infection. The asterisks indicate statistically significant differences with respect to the control without compound (* *p* < 0.05, and *** *p* < 0.001; *t*-Student) and error bars indicate standard error of the mean; *n:* 6. Moreover, are shown representative plaques of titration on VERO cells corresponding to each experimental condition.

**Figure 6 molecules-26-03430-f006:**
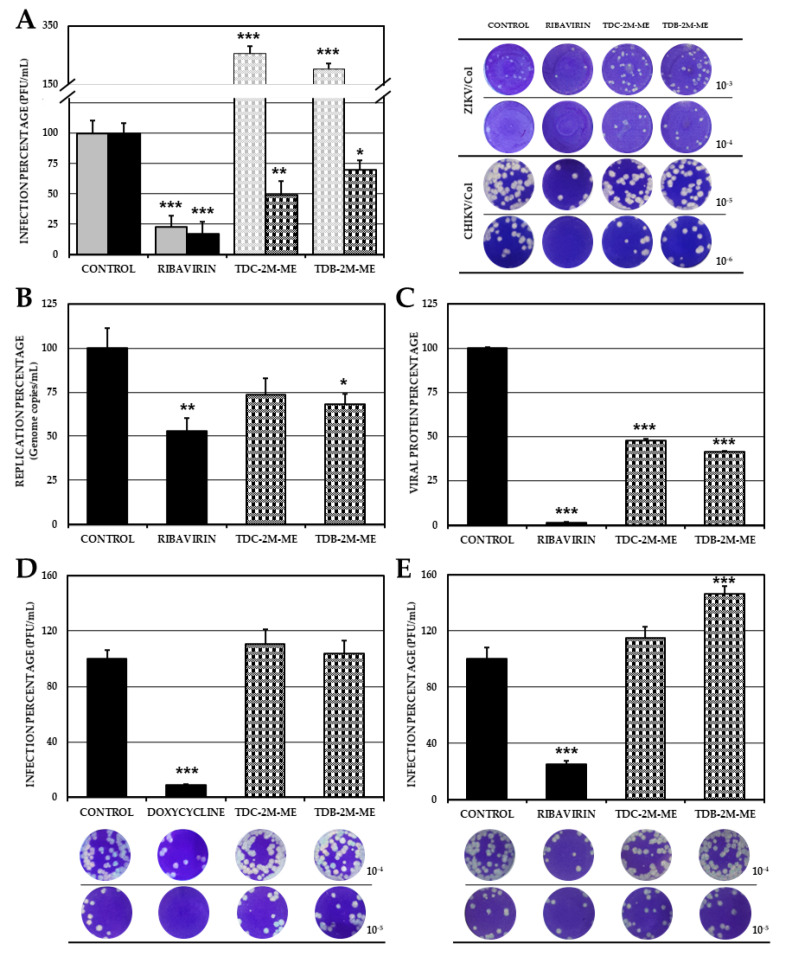
Antiviral effects of the compounds TDC-2M-ME and TDB-2M-ME in U937 cells. Infection percentages of cell cultures treated by combined strategy were calculated by plaque assay (PFU/mL) of the supernatants (**A**), real-time PCR (genome copies/mL) (**B**) and Cell-ELISA (absorbance) (**C**) in U937 monolayers. Also, the supernatants obtained of pre-treatment (**D**) and post-treatment (**E**) antiviral strategies were evaluated by plaque assay (PFU/mL). Evaluation on ZIKV/Col or CHIKV/Col infected cells are shown in gray and black, respectively. The control without treatment was assumed as 100% infection. The asterisks indicate statistically significant differences with respect to the control without compound (* *p* < 0.05, ** *p* < 0.01 and *** *p* < 0.001; *t*-Student) and error bars indicate standard error of the mean; *n:* 4. Moreover, are shown representative plaques of titration on VERO cells corresponding to each experimental condition.

**Figure 7 molecules-26-03430-f007:**
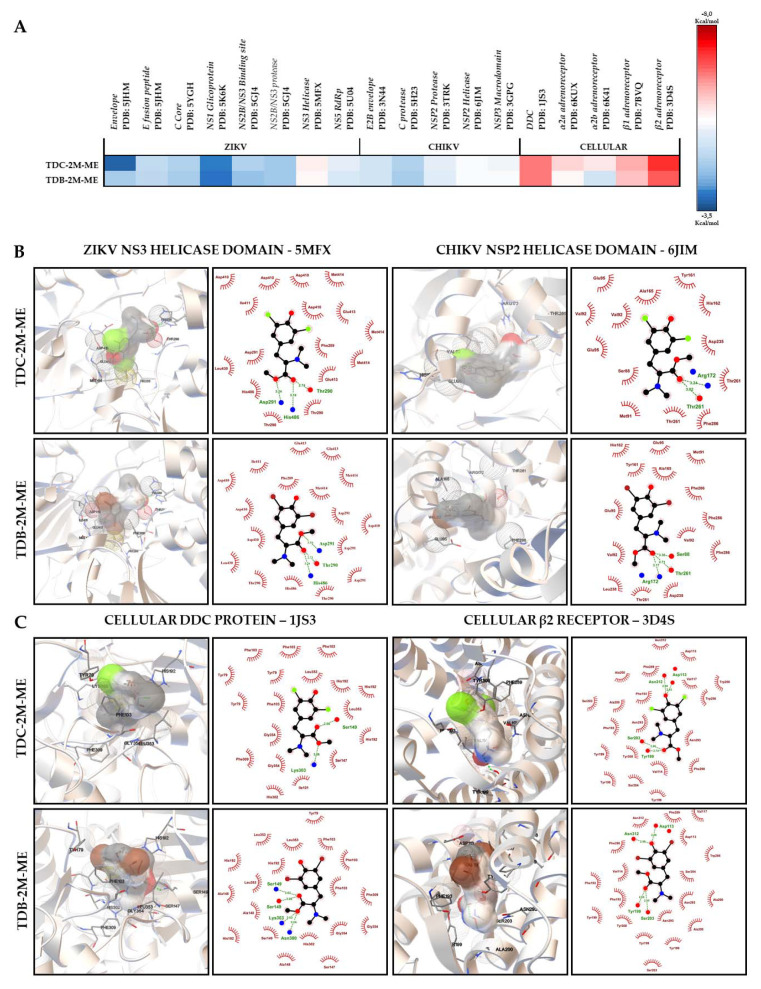
In silico interactions between viral and cellular proteins and the antiviral compounds. Heat map of binding energies. Free binding energies were obtained by molecular docking with AutodockVina^®^ of TDC-2M-ME and TDB-2M-ME compounds with viral proteins of ZIKV and CHIKV and cellular proteins related with L-tyrosine derivatives. Less than 0 kcal/mol were considered favorable energies. The averages were graphed in color scale from blue to red. Blue scale shows less favorable energies (>−6 kcal/mol), and red scale the best binding energies (≤−6 kcal/mol). The limit was established by the concordance of the results obtained from Monsalve-Escudero et al., 2021, where the in vitro (antiviral and virucidal activity) and in silico (molecular docking and molecular dynamic) results were concordant. Each complex was evaluated by triplicate (**A**). Most relevant interactions were obtained after molecular docking between the promissory antivirals and the proteins with which the best binding energies were obtained, viral helicases (**B**) and cellular proteins DDC and β2 receptor (**C**). Both graphics evidence the interactions showed by two different software PMV (left) and LigPlot^®^ (Right). Hydrogen bonds are shown in green in both software, hydrophobic interactions in red in LigPlot^®^ and π–π interactions in yellow in PMV software.

**Figure 8 molecules-26-03430-f008:**
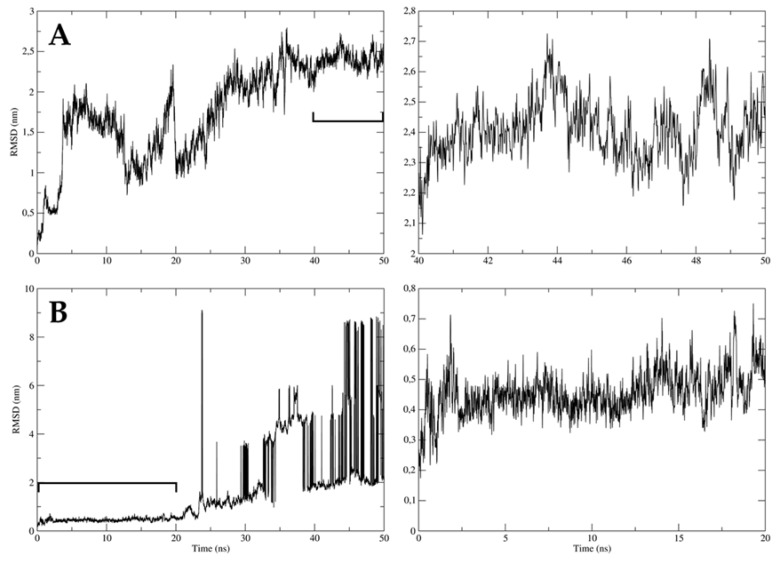
Stability of TDB-2M-ME and ZIKV envelope proteins complexes. The stability of the complexes formed by the only virucidal compound, TDB-2M-ME, and ZIKV envelope proteins, DIII (**A**) and fusion peptide (**B**), were evaluated; DIII-E ZIKV (**A**). The left plots indicate the complete simulation (50 ns) and the right plot indicates the timescale with the lowest oscillation of each simulation, 40 to 50 ns (**A**) and 0 to 20 ns (**B**). The y-axis represents the root mean square deviation (RMSD) in nanometers (nm), and the x-axis the timescale in nanoseconds (ns). The complex is considered stable if the oscillation is below 0.3 nm.

**Table 1 molecules-26-03430-t001:** Accumulated in silico toxicity score.

	Primary Amines		Tertiary Amines				Quaternary Amines			
Evaluated Toxicity	TDC	TDB	TDC-2M-ME	TDB-2M-ME	TDC-2M	TDB-2M	TDC-3M-ME	TDB-3M-ME	TDC-3M	TDB-3M
Chromosomal Aberrations	0	0	0	0	0	0	0	0	**1**	**1**
Skin Sensitization	0	**1**	0	**1**	0	**1**	0	**1**	0	**1**
Respiratory Sensitization	**1**	0	**1**	**1**	0	0	**1**	**1**	0	0
Neurotoxicity (Phospholipidosis)	0	0	0	0	0	0	**1**	0	0	0
Cardiac Toxicity	0	0	0	0	0	0	0	0	0	0
Endocrine Toxicity (estrogen receptor)	0	0	0	0	**1**	**1**	**1**	**1**	**1**	**1**
Endocrine Toxicity (androgen receptor)	0	0	0	**1**	0	0	**1**	**1**	0	0
Alkaline Phosphatase increase	0	0	**1**	0	0	0	0	0	0	0
GGT increase	0	0	0	**1**	0	**1**	0	0	0	0
LDH increase	**1**	0	**1**	0	0	0	**1**	0	0	0
SGOT increase	0	0	0	0	0	0	0	0	0	0
SGPT increase	0	0	0	0	0	0	0	0	0	0
Reproductive toxicity	0	0	0	0	0	0	0	0	0	0
**Accumulated Toxicity**	**2**	**1**	**3**	**4**	**1**	**3**	**5**	**4**	**2**	**3**

## Data Availability

The data presented in this study are available on reasonable request from the corresponding author.

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
