# Peer review of "In Vitro and In Silico Anti-Arboviral Activities of Dihalogenated Phenolic Derivates of L-Tyrosine"

_molecules, 2021, doi:10.3390/molecules26113430_

Round 1
Reviewer 1 Report
The authors have suitably addressed my comments and I have no further suggestions at this point. Minor points: Supplementary tables: Use period '.' instead of comma ',' for decimals; Throughout text and in figures: leave one decimal and not two for H-bond distances.
Author Response
REVIEWER 1
The authors have suitably addressed my comments and I have no further suggestions at this point.
- Supplementary tables: Use period '.' instead of comma ',' for decimals
Answer from author:
According to the recommendations, we changed “,” by “.” in all supplementary files.
- Throughout text and in figures: leave one decimal and not two for H-bond distances.
Answer from author:
According to the recommendations, we left only one decimal in H-bond distances (Table S4 and text).
Reviewer 2 Report
According to review’s comments, lots of revisions have been done. After reading the manuscript carefully, I think this manuscript could be published now.
Author Response
REVIEWER 2
According to review’s comments, lots of revisions have been done. After reading the manuscript carefully, I think this manuscript could be published now.
Answer from author:
Once again, we would want to thank for the previous evaluations, because they were very important to improve our manuscript.
Reviewer 3 Report
I appreciate the efforts the authors made to improve this manuscript. Most of my concerns raised has been addressed. However, the mechanism part is still not solid. In the point #6, the study about interaction between compounds and viral proteins is still confusing. The problem is that no evidence was provided to prove viral or host proteins are the targets of the inhibitors. Adding more analysis of host proteins will not address this problem. Also, even if the targets are viral proteins, the single in silico analysis can’t help make a comprehensive conclusion. Therefore, my suggestion is to add more reasonable data to support this conclusion or weaken this part and remove the description about “mechanism” in the title.
Author Response
REVIEWER 3
I appreciate the efforts the authors made to improve this manuscript. Most of my concerns raised has been addressed. However, the mechanism part is still not solid. In the point #6, the study about interaction between compounds and viral proteins is still confusing. The problem is that no evidence was provided to prove viral or host proteins are the targets of the inhibitors. Adding more analysis of host proteins will not address this problem. Also, even if the targets are viral proteins, the single in silico analysis can’t help make a comprehensive conclusion. Therefore, my suggestion is to add more reasonable data to support this conclusion or weaken this part and remove the description about “mechanism” in the title.
Answer from author:
After read carefully the actual version of the manuscript and the reviewer comment we agree and eliminated the word “mechanism” in the title. Moreover, we checked trough the manuscript and moderate our language in several sentences, to made emphasis in that our experimental strategies (In vitro and in silico) were used to try to postulate a possible mechanism, but it will be necessary to conduct other studies to define the specific antiviral mechanisms of our compounds. The changes are highlighted in green.
Round 2
Reviewer 3 Report
The revised manuscript reads well and I'm satisfied with the improvement.
This manuscript is a resubmission of an earlier submission. The following is a list of the peer review reports and author responses from that submission.
Round 1
Reviewer 1 Report
The authors report the antiviral activity of several di-halogenated compounds derived from L-tyrosine against ZIKV and CHIKV. The antiviral activity was evaluated in Vero and U937 cell lines. The potential antiviral mechanism was also investigated by molecular docking. The overall finding is potential interesting. However, the evaluation of antiviral activity is not solid and the rationale for the mechanism study is not rigorous.
- For Figures 2-5, only one concentration of the compounds was utilized to evaluate the antiviral activity. To provide more solid evidence, more concentration should be included. A dose-dependent inhibition is a golden standard for evaluation of antiviral agent.
- To systematically evaluate the antiviral potential of the di-halogenated compounds, several parameters should be included, such as IC50, CC50 and selective index (SI). The data of cytotoxicity of the tested compounds should be provided.
- For Figure 1, the absolute number of PFU/mL should be provided. The relative value is not enough to show the details.
- For Figure 2A, the copy number of viral RNA in supernatant didn’t match with the titer in the supernatant. To evaluate the viral replication, viral RNA in the infected cells should also be evaluated.
- The Cell-ELISA data is too weak to support the conclusion that viral translation is inhibited by TDB-2M-ME.
- The compounds may also target to host proteins. So the rational to study the interaction between compounds and viral proteins is not convincing.
- There are many typos in the manuscript. For example, line 110-111, “µm” should be corrected as “µM”. Line 198, “108” should be corrected as “108”.
Reviewer 2 Report
Loaiza-Cano et al. describe their characterization of the anti-viral properties of ten compounds derived from L-tyrosine, using an infected cell-line model system. They find that two compounds show promise as antivirals against ZIKV and CHIKV. The other 8 are unfortunately proviral. The authors also propose a hypothesis for what mechanism these two successful compounds would be inhibiting, and that the two successful antivrirals actually target two different pathways.
A major aspect is that 80% of the tested compounds show proviral activity, an observation which would merit further exploration. If most of these compounds increase the number of viral particles, what does that tell us about the potential effects of using such compounds as antivirals? Unaddressed, this aspect overshadows the subsequent analysis of the remaining two anti-virals (TDC-2M-ME, TDB-2M-ME), because of their chemical relatedness to the proviral compounds. It also seems that therapeutic approaches might be compromised because a compound that would be effective against one type of virus could make another one more infectious. And there would be little margin of maneuver if the slightest chemical change turns an antiviral into a proviral. The authors confuse the reader by stating (lines 337-338) that all compounds might have a therapeutic potential. Furthermore, it is not clear where these 10 compounds came from. It is alluded to in the introduction that they are from Aiolochroia crassa and Verongula rigida, but not explicitly stated.
The relevance of the in silico prediction of antiviral binding sites (section 2.7 and subsequent) is questionable, because it is unclear for example why TDC-3M-ME would "produce the best interaction" (line 266) when this compound was shown to increase and not decrease the number of infectious viral particles (Figure 1). In other words, shouldn't potential interactions only be sought out for the compounds shown to be active in Figure 1? And even for these two antivirals, are the actual protein targets known or are the authors blindly docking them as an attempt to identify targets? In any case, referring to hydrogen bonds with 2-digit decimals and number of interactions is not warranted at this point (section 2.7). These results from docking should be presented as mere testable predictions, not actual structures. Because the authors generate many hypotheses in the discussion about the function of these small molecules, to be convincing they should consider increasing the relevance of their propositions by testing binding of these compounds to the corresponding protein candidates using for example a biochemical assay, such as MST, ITC, SPR, etc. They may also want to consider testing for the enzymatic activity of the L-dopa decarboxylase as they suggest should be done (lines 334-335).
Specific comments:
- Throughout the paper statistical tests are used and p-values are partially reported, however, no standard deviation or other metric of variation is noted in the paper. Error bars are present in some of the figures but without the accompanying values. Some of the error bars seem a little high in some of the figures and it would be nice to get a handle on the variation of your replicates. Also, no actual P.Value listed just < or > 0.05.
- In the results sections, briefly explain for the sake of clarity before talking about what the combined treatment strategy, pre-treatment strategy, and post-treatment strategy are. Maybe give one sentence explanation for each of them before talking about them, as it is important to know how these strategies are employed.
- In section 2.3 it is not stated how the authors quantify the viral protein or what protein is being quantified. Is this the best way to get at this question? Also why was viral protein quantified during that experiment and not during the pre and post exposure experiments outlined in 2.4?
- In section 2.7, it would seem reasonable to propose that TDC-2M-ME and TDB-2M-ME would have similar targets in ZIKV and CHIKV. Is that what the authors are arguing for with Figure 6? The binding sites don't seem similar.
- A majority of the superscripts are missing when representing values in scientific notation.
- What would happen if all 10 compounds were tested in the U937 cell line. Would trends observed with VERO hold? Or could some other compounds now show antiviral activity in U937 but not VERO cells?
- the title of the paper would gain in being shortened
- chemical structures should be shown earlier in the manuscript
- line 50: although "most" of the time
- line 109: why not showing that data in Supplementary information instead of resorting to "data not shown"?
- line 111: a few words to explain what ADMET does exactly (pros/cons) would be helpful to the non-specialist
- line 134: missing “%” after 44.5
- line 228: “anti” is missing from viral activity
- On lines 333-335 the authors state that the cell viability was >80% at their highest concentration of 500um. This is a little out of context since they use 250 microM for all of the evaluations (section 2.1) and do not show any of the dilution series data. This data should be put into a supplemental figure to provide evidence for the rational and claim.
- lines 355-358: how do these compounds relate to the compounds studied here?
- lines 405-410: rephrase to be less vague or remove
Reviewer 3 Report
In this manuscript, Loaiza-Cano et al described the potential anti-arbovirus activity of
ten di-halogenated compounds derived from L-tyrosine with modifications in amine and carboxyl groups and studied the mechanism of action of the most promising compounds using different antiviral strategies. The results showed that none of the compounds decreased the infection of DENV, only two (TDC-2M-ME and TDB-2M-ME) inhibited ZIKV and all compounds were able to inhibit CHIKV. The manuscript is clearly organized and related to a topic of great scientific and social impact. Therefore, I recommend this manuscript to be published on Molecules after a minor revision listed below.
- The title is too complicated and needs to be simplified. It is suggested to change it to “In vitro and in silico antiviral activitiesand mechanisms of dihalogenated phenolic derivatives of L-tyrosine”.
- The chiral center has a great influence on the biological activity, so the authorshould supplement the activity data of the enantiomers of the most promising compounds.
- The format of references needs to be revised seriously.